# A silylene-stabilized ditin(0) complex and its conversion to methylditin cation and distannavinylidene

Shaozhi Du[1], Fanshu Cao[1], Xi Chen[1], Hua Rong[1], Haibin Song[1] & Zhenbo Mo [1] ✉

Due to their intrinsic high reactivity, isolation of tin(0) complexes remains challenging. Herein, we report the synthesis of a silylene-stabilized ditin(0) complex (**2**) by reduction of a silylene-supported dibromostannylene (**1**) with 1 equivalent of magnesium (I) dimer in toluene. The structure of **2** was established by single crystal X-ray diffraction analysis. Density Functional Theory calculations revealed that complex **2** bears a Sn=Sn double bond and one lone pair of electrons on each of the Sn(0) atoms. Remarkably, complex **2** is readily methylated to give a mixed-valent methylditin cation (**4**), which undergoes topomerization in solution though a reversible 1,2-Me migration along a Sn=Sn bond. Computational studies showed that the three-coordinate Sn atom in **4** is the dominant electrophilic center, and allows for facile reaction with KHBBu$^s_3$ furnishing an unprecedented N-heterocyclic silylenes-stabilized distannavinylidene (**5**). The synthesis of **2**, **4** and **5** demonstrates the exceptional ability of N-heterocyclic silylenes to stabilize low valent tin complexes.

The stabilization of main group elements in unusual oxidation states is an important topic in main group chemistry[1–4]. Since the landmark report of the isolation of a disilicon(0) complex by employing an N-heterocyclic carbene (NHC)[5], a number of zero-valent diatomic main-group compounds stabilized by NHCs have been prepared (Fig. 1a)[6–21]. With their unique electronic features and their synthetic potential, their reactivity studies are of fundamental importance[22–27]. It is particularly interesting that NHC-stabilized zero-valent diatomic main group compounds have been shown to activate small molecules and serve as precursors for the construction of long sought after molecules that would otherwise remain inaccessible[22–27]. As notable examples, disilicon(0) and diphosphorus(0) complexes can be directly oxidized by O$_2$ to give the NHC-stabilized silicon oxide (Si$_2$O$_4$) and phosphorus oxide (P$_2$O$_4$)[28–30], and the reactions of diboron(0) complexes with CO afforded the bis(boralactone) and bis(boraketene) compounds[31,32]. Recently, a 1,2-diboraallen was used as a precursor for the synthesis of a novel tetraatomic boron(0) species, a planar tetraatomic boron(0) unit with the average oxidation state of zero[33].

Despite these advances, access to zero-valent diatomic main-group compounds is still in the early stages and remains a challenging research subject. On account of its low electronegativity and the less effective pπ–pπ orbital overlap of the tin atom, the chemistry of ditin(0) compounds has progressed relatively slowly. In 2012, Jones et al. reported the isolation of a ditin(0) complex bonded by an NHC ligand[11], which is the only example of a structurally characterized ditin(0) complex. However, studies of its reactivity have been elusive probably due to its tendency to decompose at room temperature both in solution and in the solid state. Therefore, the development of new stabilization strategies to access bottleable ditin(0) complexes will be important in the elucidation of their chemical reactivity. Recently, N-heterocyclic silylenes (NHSi), heavier NHC analogs, has been proved to be capable of stabilizing zero-valent diatomic main group compounds including disilicon(0)[34], diphosphorus(0)[35] and digermanium(0)[36] complexes. Theoretical studies have demonstrated that NHSi-stabilized ditin(0) complexes are synthetically achievable[37], but their direct observation to date has not been reported. In this context, we report herein the synthesis and characterization of an NHSi-ligated ditin(0) complex (**2**) and its conversion to a mixed-valent methylditin cation (**4**) and distannavinylidene compound (**5**) (Fig. 1b). The electronic structures of **2**, **4** and **5** have been investigated by computational methods.

[1]State Key Laboratory and Institute of Elemento-Organic Chemistry, Frontiers Science Center for New Organic Matter, College of Chemistry, Nankai University, Tianjin, China. ✉e-mail: zhenbo.mo@nankai.edu.cn

### (a) NHC-stabilized Zero-Valent Diatomic Main-Group Compounds

E = C, Si, Ge, Sn                                  Pn = P, As, Sb

### (b) NHSi-Stabilized Ditin(0), Methylditin Cation and Distannavinylidene

***Ditin(0)***          ***Methylditin Cation***          ***Distannavinylidene***

**Fig. 1 | Zero-valent diatomic main-group compounds stabilized by NHC and NHSi ligands. a** NHC-stabilized zero-valent diatomic main-group compounds. **b** NHSi-stabilized ditin(0), methylditin cation, and distannavinylidene.

## Results

The LSi(NHI) adduct of dibromostannylene [SnBr$_2${LSi(NHI)}] (**1**, L = PhC(N$^t$Bu)$_2$, NHI = 2,6-diisopropylphenyl-imidazoline-2-imino) was facially synthesized from the ligand substitution reaction of an *N*-heterocyclic imino substituted silylene with [(IPr)SnBr$_2$] (IPr = 1,3-bis(2,6-diisopropylphenyl)-imidazol-2-ylidene) in diethyl ether. Complex **1** was isolated in 90% yield as yellow crystals and its structure was confirmed by X-ray crystallography. The Si−Sn bond length in 1 (2.725(1)Å) is comparable to that in a bis(NHSi)-supported dibromo-stannylene (2.709 Å)[38]. Treatment of **1** with 1 equiv of Jones' magnesium(I)-reducing agent[39] [($^{Mes}$Nacnac)Mg]$_2$ in toluene immediately produced a green solution, from which the NHSi-stabilized ditin(0) complex [{L(NHI)Si}Sn = Sn{Si(NHI)L}] (**2**) was isolated in 60% yield after workup (Fig. 2). In contrast to the thermally sensitive NHC-supported ditin(0) complex, complex **2** is stable in toluene at ambient temperature for several days. Complex **2** can also be synthesized by the reaction of KHBBu$^s$$_3$ with an NHSi-stabilized ditin(I) bromide, [{L(NHI)Si}Sn(Br) = Sn(Br){Si(NHI)L}] (**3**) that is obtained from the reduction of **1** with 0.5 equiv of [($^{Mes}$Nacnac)Mg]$_2$ (Fig. 2). Complex **2** was characterized by NMR spectroscopy and single-crystal X-ray diffraction analysis. The $^{29}$Si NMR spectrum of **2** shows one singlet at δ = −5.89 ppm which is shifted downfield relative to the NMR signal of 1 (−21.19 ppm). The $^{119}$Sn NMR resonances were not observed for **2** in the range from −3000 to 3000 ppm, most likely due to the anisotropy of the shift tensor. The UV/Vis absorption spectrum of **2** measured from its solution in benzene has one absorption band at ~587 nm, which is assigned to the HOMO → LUMO transition as revealed by the time-dependent DFT (TD-DFT) calculations (see Supplementary Information).

Single crystals of **2** suitable for X-ray diffraction were grown by cooling a saturated solution of **2** in THF at −30 °C. The X-ray crystal structure of **2** (Fig. 3) shows that the central SiSnSnSi motif is trans-bent with a torsion angle Si1-Sn1-Sn1A-Si1A of 180.00(3)°, which resembles those of the isolobal diphosphene Mes*P = PMes*[40] (Mes* = 2,4,6-$^t$Bu$_3$C$_6$H$_2$), distibene [(Ar$^{Me6}$)Sb = Sb (Ar$^{Me6}$)][41] (Ar$^{Me6}$ = 2,4,6-Me$_3$-C$_6$H$_2$)[41] and phosphasilenylidene (IPr)Si = PMes*[42]. The Sn1−Sn1A distance (2.7240(6) Å) in **2** is comparable to that in the NHC-stabilized ditin(0) complex (2.7225(5) Å)[11], but is longer than the corresponding

bond in the distannyne [Ar'Sn≡SnAr'] (Ar' = C$_6$H$_3$−2,6-Dipp$_2$, 2.6675(4) Å)[43]. The Si1–Sn1 bond length of 2.607(1) Å is significantly shorter than the corresponding bond in complex **1** (2.725(1) Å) as a result of the strong electron delocalization from the Sn−Sn σ and π orbitals to the formal empty p-orbital of the silicon atom as revealed by DFT calculations.

To gain insight into the nature of the bonding, DFT calculations were performed at the PBE0/Def2SVP level with Grimme's D3BJ dispersion correction[44–46]. The optimized structure of **2** agrees well with the X-ray derived structure (Supplementary Table. 3) with the overall deviation of ~1%. The highest occupied molecular orbital (HOMO) is the π(Sn−Sn) orbital (Fig. 4), while the lowest unoccupied molecular orbital (LUMO) represents the π*(Sn−Sn) orbital (Supplementary Fig. 47) and the HOMO − LUMO gap is 2.48 eV. The HOMO − 1 displays the σ (Sn−Sn), and the lone pairs at Sn atoms can be found in the HOMO-2 and HOMO-3. For comparison purposes, the electronic structure of an isolobal compound MeSb = SbMe was studied by DFT calculations at the same level of theory. Its frontier orbitals have same number and symmetry properties as those of **2**, that is, the HOMO is dominated by the π(Sb−Sb) orbital and the LUMO is the π*(Sb−Sb) orbital (Supplementary Fig. 49). Natural bond orbital (NBO) analysis suggests that the Wiberg bond index (WBI) of the Sn−Sn bond in **2** is 1.77, which is consistent with its double bond character. Both the π(Sn−Sn) and σ(Sn−Sn) enjoy considerable delocalization as evidenced by the low Lewis occupancies of 1.80 and 1.89 e$^-$, respectively. A second-order perturbation analysis confirmed donor−acceptor interactions between the donor π(Sn−Sn) and σ(Sn−Sn) NBOs and the acceptor LV-NBO at the two Si atoms with stabilization energies of 18.32 and 33.91 kcal mol$^{-1}$, respectively. The natural population analysis (NPA) charges in **2** (Sn: −0.49; Si: 1.49) reflect the donor−acceptor character of the Si−Sn bond, which is composed of a sp$^{0.67}$ hybrid orbital at the Si atom (66%) and a p orbital at the Sn atom (34%). With these facts, the structure of **2** can be most accurately described as a NHSi-stabilized ditin(0) complex.

It is anticipated that complex **2** might be serve as a Sn(0) atom transfer reagent on account of its good solubility. The reaction of complex **2** with four equivalents of DippN$_3$ in Et$_2$O proceeded immediately along with effervescence of dinitrogen (Fig. 5). The resulting $^1$H,

**Fig. 2 | Preparation of the NHSi-stabilized ditin(0) complex (2).** The reaction of **1** with 1 equivalent of Jones' magnesium(I)-reducing agent in toluene affording ditin (0) complex (**2**). The reaction of **1** with 0.5 equivalent of Jones' magnesium(I)-reducing agent in toluene affording ditin(I) bromide (**3**). The reaction of **3** with 2 equivalents of KHBBu$^s_3$ in THF affording ditin (0) complex (**2**).

[13]C and [119]Sn NMR spectrum revealed the formation of the tetrameric tin imido complex [(SnNDipp)$_4$], which has previously been synthesized by the reaction of Sn[N(SiMe$_3$)$_2$]$_2$ with DippNH$_2$[47]. The reaction is thought to proceed through the oxidation of Sn(0) by DippN$_3$ to produce a tin imide species, which subsequently undergoes tetramerization to give [(SnNDipp)$_4$]. In addition, treatment of **2** with six equivalents of imQ (4,6-di-tert-butyl-N-(2,6-diisopropylphenyl)-o-imi-nobenzoquinone) gave a colorless solution, from which the five-coordinate bis(amidophenolato)tin(IV) complex was isolated[48] (Fig. 5). In these reactions, the dissociated silylene ligands further reacted with DippN$_3$ and imQ to give the oxidation byproducts (see Supplementary Information). These reactions demonstrated the ability of the silylene-stabilized ditin(0) complex in delivering its Sn(0) atom to other substrates.

Given the electron rich property of **2**, we examined the reactions with simple electrophiles such as H$^+$ and Me$^+$ with the aim to prepare ditin cation species. The synthesis of such species is challenging as they possess highly reactive main-group element multiple bonds and cations. Recently, Filippou et al. reported the preparation of intri-guing disilicon cations via protonation and alkylation of [(IPr)Si = Si(IPr)][49,50]. The reaction of **2** with 1 equiv of [H(Et$_2$O)$_2$][B(Ar$^F$)$_4$] gave an inseparable mixtures. However, the reaction of **2** with 1 equiv of methyl triflate (MeOTf) in the presence of Na[B(Ar$^F$)$_4$] (1 equiv, Ar$^F$ = 3,5-(CF$_3$)$_2$-C$_6$H$_3$) in fluorobenzene furnished a purple solution, from which a NHSi-stabilized mixed-valent methylditin cation [{L(NHI)Si}Sn(Me) = Sn{Si(NHI)L}][B(Ar$^F$)$_4$] (**4**) was isolated as purple crystals in 90% yield (Fig. 6). The [1]H NMR spectrum of **4** in C$_6$D$_6$ shows one singlet at δ = 1.52 ppm which is attributed to the Sn−CH$_3$ protons, and a single resonance at δ = 6.00 ppm from the CH groups of

imidazole rings, revealing the fast exchange of the two Sn sites involving a 1,2-Me migration in solution on the NMR time scale. The variable-temperature NMR studies on the solutions of **4** in d$_8$-toluene show that decreasing the temperature to 213 K led to the splitting of the single set of signals for NHI group into a double set of signals. The [119]Sn NMR spectrum of **4** recorded at 213 K shows two signals at δ 633.4 and 297.9 ppm, which correspond to the triicoordinate Sn(II) atom and the terminal two-coordinated Sn(0) atom, respectively. The resonance for the Sn(0) atom is shifted downfield relative to that of the bis(NHSi)-stabilized zero-valent tin complex (δ = −1147.2 ppm)[38]. To gain insight into the mechanism of the topomerization of **4**, we performed DFT calculations employing a model system at the PBE0-D3(BJ)/Def2-TZVP level of theory. DFT studies show that the 1,2-Me migration along the Sn = Sn bond proceeds via a NHSi-stabilized distannamethonium ion. The migration of the end-on bonded methyl group in **4Me** to a bridging position yields a dis-tannamethonium ion (**4Me'**) formed via **4Me-TS** with a small free energy barrier of 6.8 kcal mol$^{-1}$ (Supplementary Fig. 51). In **4Me'**, the Sn−Sn bond (2.82 Å) is elongated relative to the same bond in 4Me (2.69 Å). The Sn−C bond distances in **4Me'** are however markedly different (2.34 and 2.76 Å), which are both longer than that the cor-responding bond in **4Me** (2.18 Å).

An X-ray diffraction study established that **4** possesses a trans-bent geometry with a Si(1)Sn(1)Si(2)Sn(2) torsion angle of 172.0(3)° (Fig. 7). Significantly, the Sn1 − Sn2 − Si2 (96.2(2)°) bond angle is sig-nificantly smaller than the Si1 − Sn1 − Sn2 bond angle (113.2(2) °), which suggests the presence of a lone pair of electrons at the Sn2 atom. The Si1 − Sn1 bond length of 2.512(7) Å is shorter than that of Si2 − Sn2 (2.630(8) Å) as a result of the electron deficiency of the Sn1 atom. The

Sn1 − Sn2 bond distance in **4** (2.646(1) Å) is about 0.08 Å shorter than that in **2**, which might be caused by the decreased lone-pair repulsion. The Sn−C bond distance in **4** (2.186(8) Å) is comparable to that in an aryl-substituted Sn(I) radical (2.2038(17) Å)[51].

The methylditin cation **4** is isolobal to multiple-bonded phosphorous compounds such as diphosphanyl cations (Fig. 8, A) and stannaphosphenes (Fig. 8, B). These structures display a trigonal planar geometry at the three-coordinated Sn and P atom. Recent quantum chemical calculations of [Me₂P = PMe]⁺ and [(Ar^Dipp){CH₂P(CH₃)₂}Sn = P(Ar^Dipp)] (Ar^Dipp = 2,6-(2,6-^iPr-C₆H₃)-C₆H₃) by the Filippou group[49] and

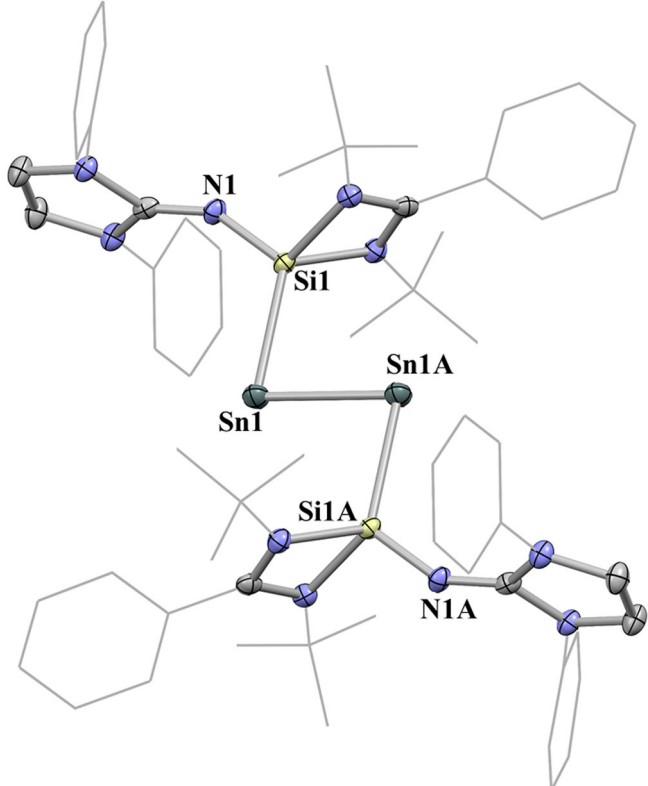

**Fig. 3 | The molecular structure of complex 2 showing 30% probability ellipsoids.** The ^iPr groups and hydrogen atoms have been omitted for clarity. Selected bond lengths (Å) and angles (º): Sn1 − Sn1A 2.7240(6), Si1 − Sn1 2.607(1), Si1 − N1 1.673(3), Sn1 − Si1 − N1 121.1(1), Si1 − Sn1 − Sn1A 89.20(3).

the Aldridge group[52], respectively, reveal that they have similar frontier orbitals as those of **4**. The HOMO can be identified as the π(E − E) orbital (E = Sn or P) in all cases and the LUMO represents the π*(E − E) orbitals. The nature of the bonding in **4** was further studied by NBO calculations (PBE0-D3(BJ)/Def2-SVP), which show that the lone pair of electrons at Sn2 atom features a high (85%) s-character and the π(Sn−Sn) orbital is formed from p-orbitals of the two Sn atoms with an occupancy of 1.86 e⁻ (Supplementary Fig. 52). The WBI of the Sn−Sn bond in 4 (1.84) is larger than that of **2** and is in line with its enhanced double bond character. To obtain a better understanding of the charge distribution of **4**, the natural population analysis (NPA) charges were examined. The Sn1 atom bears a + 0.24 positive charge and the Sn2 atom has a −0.24 negative charge, which shows that the electron transfer from **2** to the electrophile (q(Me) = −0.41) is mainly from the Sn1 atom.

According to the computational findings, the positive charge in **4** is mainly at the Sn1 atom. Thus, we envisioned that the addition of nucleophiles to **4** might produce the tin analogs of vinylidene, which remain unknown notwithstanding the considerable efforts that have been made over the past decade to synthesize heavier analogs of vinylidene compounds[53–59]. Our initial attempts to prepare the silylene-supported analogs via the reactions of **4** with alkyl and aryl Grignard and lithium reagents gave complex mixtures. Interestingly, when **4** was treated with KHBBu^s₃ in THF, a purple solution was obtained, from which a distannavinylidene (**5**) with a silyl and a methyl substituent was isolated as purple crystals in 30% yield (Fig. 9). Distannavinylidene **5** might be formed via two different routes. One direct pathway proceeds by the addition of the hydride to the PhC site of amidinate as the LUMO + 1 of **4** is comprised of the π* orbitals localized on the amidinate. An alternative pathway involves the addition of the hydride to the Sn1 atom followed by the cleavage of the Sn−H bond by NHSi in a cooperative fashion, as has been recently described in the cooperative B−H bond activation by the amidinate-stabilized silylene.[60] Compound **5** was characterized by NMR spectroscopy, UV/Vis-NIR spectroscopy, single-crystal X-ray diffraction and elemental analysis. The ¹H NMR spectrum of **5** recorded in C₆D₆ displays one singlet at δ = 5.51 ppm from the CHPh methylene group. The ²⁹Si NMR spectrum of **5** also shows two non-equivalent resonances at −28.83 and −7.40 ppm. In the ¹¹⁹Sn NMR spectrum, the peak at δ 843.2 ppm, which corresponds to the tri-coordinated tin atom, is shifted downfield relative to that of **1** (−21.3 ppm), and the other signal at δ 300.3 ppm, assigned to the terminal two-coordinated tin atom, is comparable to that of the Sn(0) atom in **4**. The UV−Vis absorption spectrum of **5** in THF has two broad

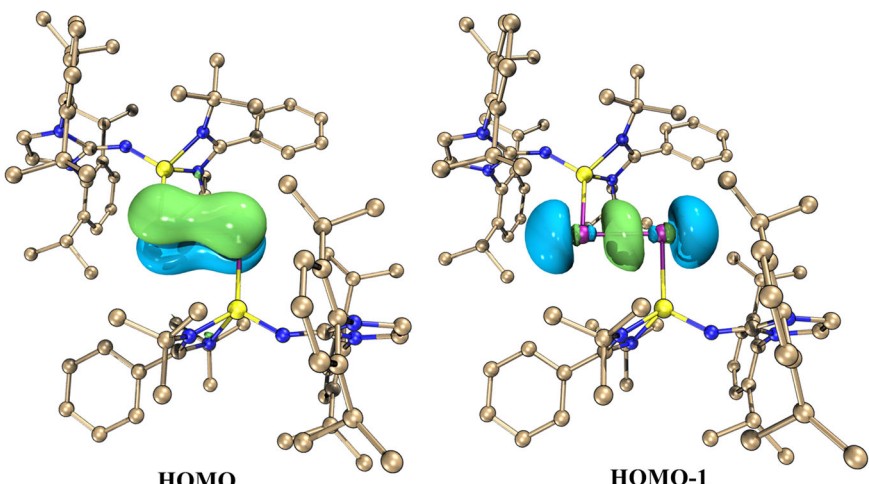

**Fig. 4 | Selected frontier orbitals of 2 determined by DFT calculations (isovalue = 0.05).** The HOMO and HOMO−1 of complex **2** represent the π(Sn−Sn) orbital and σ(Sn−Sn) orbital, respectively.

**Fig. 5 | Transferring the Sn atom to DippN₃ and imQ to give the tetrameric imido complex and Tin (IV) amidophenolate complex.** The reaction of **2** with 4 equivalents of DippN₃ leading to tetrameric imido complex and the reaction of **2** with 6 equivalents of imQ leading to tin (IV) amidophenolate complex.

**Fig. 6 | Preparation of the NHSi-stabilized mixed-valent methylditin cation (4).** The reaction **2** with 1 equivalent of methyl triflate in the presence of Na[BArᶠ₄] in fluorobenzene affording the complex **(4)**.

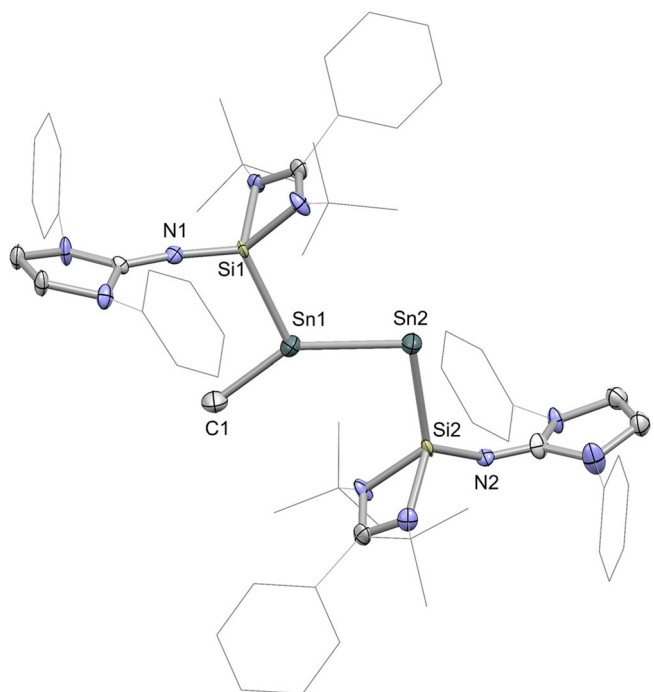

**Fig. 7 | The molecular structure of complex 4 showing 30% probability ellipsoids.** The ⁱPr groups and hydrogen atoms have been omitted for clarity. Selected bond lengths (Å) and angles (°): Sn1 – Sn2 2.646(1), Si1 – Sn1 2.512(7), Si2 – Sn2 2.630(8), Sn1 – C1 2.186(8), Si1 – Sn1 – Sn2 113.2(2), Sn1 – Sn2 – Si2 96.2(2), Si1 – Sn1 – C1 105.5(3), C1 – Sn1 – Sn2 140.8(2), Si1 – Sn1 – Sn2 – Si2 172.0(3).

absorption bands at ~396 and ~533 nm. The maximum band at 533 nm is close to the value calculated for the $\pi_{Sn=Sn} \rightarrow \pi^*_{Sn=Sn}$ transition (see Supplementary information).

X-ray crystallography has shown that compound **5** is a NHSi-stabilized distannavinylidene (Fig. 10). The H1 atom was detected in the electron density map. Accordingly, the bond distances of C2 – N3 (1.46(2) Å) and C2 – N4 (1.48 (2) Å) are much longer than those of C3 – N5 and C3 – N6 (1.35(2) and 1.40 (1) Å). The Sn1 atom in **5** is tricoordinated to one Si atom, one C atom and one Sn atom with a sum of the angles of 359.95°. The Sn2 atom features a V-shaped geometry with a Sn1 – Sn2 – Si2 angle of 97.79(9)°. The Sn1–Sn2 distance (2.654(1) Å) in **5** is comparable to that in **4** (2.646(1) Å), which is consistent with its double bond character.

DFT calculations coupled with NBO analysis provide a thorough insight into the nature of the Sn1–Sn2 bonding in **5**. The HOMO of **5** comprises the Sn1–Sn2 π orbital, where the HOMO-2 is the lone pair of electrons at the Sn2 atom (Supplementary Fig. 53). Complex **5** possesses same sequence of frontier orbitals as those of **2**, diphosphanyl cation and stannaphosphene, which is in accordance with their isolobal relationship (Fig. 8). NBO analysis shows that the Sn1–Sn2 σ bond is polarized toward the Sn1 atom (60%) with an occupancy of 1.92 e⁻ and the depleted Sn1–Sn2 π bond, with an occupancy of 1.85 e⁻, results from the electron delocalization to the formally empty p(Si) orbitals of NHSi (Supplementary Fig. 54). The calculated Wiberg bond index (WBI) value of Sn1–Sn2 (1.32) is consistent with its weak double bond character.

## Discussion

In summary, by using a strong electron-donating and sterically hindered N-heterocyclic imino substituted silylene, we isolated a NHSi-stabilized ditin(0) complex **(2)**, which is stable at rt both in solution and in the solid state. Theoretical studies demonstrated that **2** features a Sn(0) = Sn(0) double bond with partial delocalization of electrons into the formal empty p(Si) orbitals of NHSis. In addition, the ditin(0) compound **(2)** can be used as a precursor for the preparation of the NHSi-stabilized mixed-valent methylditin cation **(4)** and distannavinylidene **(5)**, which are characterized by various methods. The Sn⁰ atom transfer reactions of ditin(0) complex **(2)** further highlight its synthetic utility. This work demonstrates the utility of NHSis in stabilization of tin(0) complexes and provides a strategy to probe their exciting chemistry.

**Fig. 8 | Isolobal phosphorous derivatives of complexes 4 and 5, R = anionic ligands, Silyl = [Ph(H)C(N$^t$Bu)$_2$]Si.** Representative multiple-bonded phosphorous compounds such as diphosphanyl cations (Fig. 8, A) and stannaphosphenes (Fig. 8, B), which are isolobal to complexes **4** and **5**.

**Fig. 9 | Preparation of the NHSi-stabilized distannavinylidene (5).** The reaction of **4** with 1 equivalent of KHBBu$^s_3$ in THF affording the distannavinylidene (**5**).

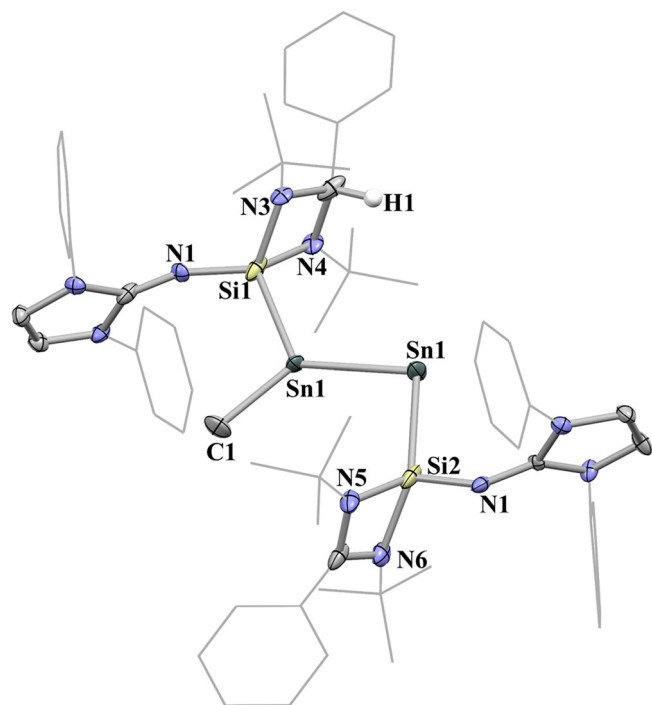

**Fig. 10 | The molecular structure of complex 5 showing 30% probability ellipsoids.** The $^i$Pr groups and hydrogen atoms have been omitted for clarity. Selected bond lengths (Å) and angles (°): Sn1 − Sn2 2.654(1), Si1 − Sn1 2.537(4), Si2 − Sn2 2.676(4), Sn1 − C1 2.16(2), Si1 − N1 1.67(1), Si2 − N2 1.64(1), C2 − H1 1.00, C2 − N3 1.46(2), C2 − N4 1.48(2), C3 − N5 1.35(2), C3 − N6 1.40(1), Si1 − Sn1 − Sn2 119.04(9). Sn1 − Sn2 − Si2 97.79(9), Si1 − Sn1 − C1 110.9(4), C1 − Sn1 − Sn2 130.0(4), Si1 − Sn1 − Sn2 − Si2 175.7(1).

## Methods
### General procedures
All reactions were carried out under a dry and oxygen-free argon or dinitrogen atmosphere by using Schlenk techniques or under an argon

or dinitrogen atmosphere in a Vigor glovebox. The argon or dinitrogen in the glovebox was constantly circulated through a copper/molecular sieves catalyst unit. The oxygen and moisture concentrations in the glovebox atmosphere were monitored by an O$_2$/H$_2$O Combi-Analyzer (Vigor LG2400/750TS-F) to ensure both were always below 1 ppm. Full information to the synthetic, spectroscopic, crystallographic, and computational methods is given in Supplementary Information.

**Preparation of Compound 1:** A Et$_2$O solution (5 mL) of LSi(NHI) (L = PhC(N$^t$Bu)$_2$) (662 mg, 1.0 mmol) was added to a stirred suspension of IPr→SnBr$_2$ (667 mg, 1.0 mmol) in Et$_2$O (5.0 mL) dropwise at room temperature. After stirring for 2 h, a yellowish slurry was obtained. The yellowish precipitate was isolated by filtration in 90 % yield. Single crystals suitable for X-ray diffraction studies were obtained by vapor diffusion of n-hexane into the THF solution at room temperature.

$^1$H NMR (400 MHz, THF-$d_8$): δ 7.97 (d, $J$ = 7.2 Hz, 1H, Ar$H$), 7.47-7.41 (m, 2H, Ar$H$), 7.39-7.35 (m, 4H, Ar$H$), 7.30-7.28 (m, 3H, Ar$H$), 7.17 (d, $J$ = 7.6 Hz, 1H, Ar$H$), 6.73 (s, 2H, NC$H$), 3.14-3.04 (m, 4H, C$H$(CH$_3$)$_2$), 1.43 (d, $J$ = 7.2 Hz, 12H, CH(C$H_3$)$_2$), 1.22 (d, $J$ = 6.8 Hz, 12H, CH(C$H_3$)$_2$), 0.74 (s, 18H, C(C$H_3$)$_3$).

$^{13}$C{$^1$H} NMR (101 MHz, THF-$d_8$, 298 K): δ 173.4 (s, N$C$N), 148.1 (s, Ar$C$), 144.1 (s, Ar$C$), 135.2 (s, Ar$C$), 133.5 (s, Ar$C$), 132.1 (s, Ar$C$), 131.3 (s, Ar$C$), 130.6 (s, Ar$C$), 129.3 (s, Ar$C$), 128.8 (s, Ar$C$), 128.4 (s, Ar$C$), 125.4 (s, Ar$C$), 117.3 (s, N$C$H), 54.24 (s, $C$(CH$_3$)$_3$), 31.52 (s, $C$H(CH$_3$)$_2$), 29.66 (s, $C$H$_3$), 24.52 (s, $C$H$_3$). one peak for $C$H$_3$ are overlapped with the solvent residual signal of THF-$d_8$.

$^{29}$Si{$^1$H} NMR (79 MHz, THF-$d_8$, 298 K): δ −21.19 (s).

$^{119}$Sn{1H} NMR (149 MHz, THF-$d_8$, 298 K): δ 181.6 ($^1J_{Si, Sn}$ = 1245 Hz).

Anal. Calcd for C$_{42}$H$_{59}$Br$_2$N$_5$SiSn: C, 53.63; H, 6.32; N, 7.45. Found: C, 53.27; H, 6.43; N, 7.63.

**Preparation of Compound 2:**

**Method A:** A toluene solution (10 mL) of {($^{Mes}$Nacnac)Mg}$_2$ (715.6 mg, 1.0 mmol) was added to a stirred suspension of 2 (940.6 mg, 1.0 mmol) in toluene (5.0 mL) dropwise at −30 °C. The mixture was allowed to warm to room temperature and stirred for 4 h. Then the solvent was removed under vacuum and the resulting solids were extracted with 30 mL THF. After filtration and removal of the solvent, the resulted brown solid was washed with 60 mL Et$_2$O to yield dark green powder of **2** in 60% yield. Single crystals suitable for

X-ray diffraction studies were grown by cooling the THF solution at −30 °C.

**Method B:** A THF solution (1.0 mL) of KHB$^s$Bu$_3$ (221.3 mg, 1.0 mmol) was added to the **3** (861 mg, 0.5 mmol) in THF (5.0 mL) dropwise at room temperature over a period of 2 min. The mixture was stirred for another 5 min. After filtration and removal of the solvent, the resulting brown solid was washed with 5 mL Et$_2$O. Then the resulting solid was dissolved in a mixture of THF (10 mL) and *n*-hexane (5 mL). The suspension was filtered and the filtrate was stored at −30 °C for 2 days to yield the green powder of **2** in 40% yield.

$^1$H NMR (400 MHz, THF-d$_8$): δ 8.54 (d, *J* = 6.8 Hz, 2H, Ar*H*), 7.39-7.29 (m, 7H, Ar*H*), 7.27−7.22 (m, 7H, Ar*H*), 7.16−7.15 (m, 6H, Ar*H*), 6.43 (s, 4H, NC*H*), 3.33−3.24 (m, 8H, C*H*(CH$_3$)$_2$), 1.34 (d, *J* = 6.4 Hz, 24H, CH(C*H*$_3$)$_2$), 1.16 (d, *J* = 6.8 Hz, 24H, CH(C*H*$_3$)$_2$), 0.72 (s, 36H, C(C*H*$_3$)$_3$).The solubility in THF-d$_8$ is not good enough.

$^{13}$C{$^1$H} NMR (101 MHz, THF-d$_8$, 298 K): δ 170.1 (s, N*C*N), 149.1 (s, Ar*C*), 145.7 (s, Ar*C*), 137.6 (s, Ar*C*), 136.6 (s, Ar*C*), 135.0 (s, Ar*C*), 130.1 (s, Ar*C*), 129.8 (s, Ar*C*), 129.7(s, Ar*C*), 127.8 (s, Ar*C*), 127.7 (s, Ar*C*), 124.8 (s, Ar*C*), 116.6 (s, N*C*H), 54.16 (s, *C*(CH$_3$)$_3$), 32.03 (s, *C*H(CH$_3$)$_2$), 29.54 (s, *C*H$_3$), 24.76 (s, *C*H$_3$). one peak for *C*H$_3$ are overlapped with the solvent residual signal of THF-d$_8$.

$^{29}$Si{$^1$H} NMR (79 MHz, THF-d$_8$, 298 K): δ −5.89 (s).

$^{119}$Sn{$^1$H} NMR (149 MHz, THF-d$_8$, 298 K): no signal was observed in the range from -3000 to 3000 ppm, most likely due to the anisotropy of the shift tensor.

Absorption spectrum (THF): λmax (ε) = 587 (8713), 459(5715) and 264(78513) nm.

Anal. Calcd for C$_{84}$H$_{118}$N$_{10}$Si$_2$Sn$_2$: C, 64.61; H, 7.62; N, 8.97. Found: C, 64.81; H, 7.75; N, 8.95.

**Preparation of Compound 3:** A toluene solution (10 mL) of {($^{Mes}$Nacnac)Mg}$_2$ (357.8 mg, 0.5 mmol) was added to a stirred suspension of **2** (940.6 mg, 1.0 mmol) in toluene (5.0 mL) dropwise at −30 °C. The mixture was allowed to warm to room temperature and stirred for 0.5 h. After removal of the solvent, the resulted solid was washed with 30 mL Et$_2$O to yield green powder of **3** in 90 % yield. Single crystals suitable for X-ray diffraction studies were obtained by vapor diffusion of *n*-hexane into the benzene solution at room temperature.

$^1$H NMR (400 MHz, THF-d$_8$): δ 8.15 (d, *J* = 7.2 Hz, 2H, Ar*H*), 7.39−7.28 (m, 12H, Ar*H*), 7.25−7.23 (m, 6H, Ar*H*), 7.14 (d, *J* = 8.0 Hz, 2H, Ar*H*), 6.84 (s, 4H, NC*H*), 3.31−3.21 (m, 8H, C*H*(CH$_3$)$_2$), 1.47 (d, *J* = 6.8 Hz, 24H, CH(C*H*$_3$)$_2$), 1.18 (d, *J* = 6.8 Hz, 24H, CH(C*H*$_3$)$_2$), 0.79 (s, 36H, C(C*H*$_3$)$_3$).

$^{13}$C{$^1$H} NMR (101 MHz, THF-d$_8$, 298 K): δ 171.2 (s, N*C*N), 148.8 (s, Ar*C*), 144.1 (s, Ar*C*), 136.2 (s, Ar*C*), 133.2 (s, Ar*C*), 130.2 (s, Ar*C*), 129.71 (s, Ar*C*), 129.69 (s, Ar*C*), 128.9 (s, Ar*C*), 128.1 (s, Ar*C*), 127.5 (s, Ar*C*), 124.9 (s, Ar*C*), 116.8 (s, N*C*H), 53.70 (s, *C*(CH$_3$)$_3$), 31.74 (s, *C*H(CH$_3$)$_2$), 29.31 (s, *C*H$_3$), 25.18 (s, *C*H$_3$). one peak for *C*H$_3$ is overlapped with the solvent residual signal of THF-d$_8$.

$^{29}$Si{$^1$H} NMR (79 MHz, THF-d$_8$, 298 K): δ −23.29 (s).

$^{119}$Sn{$^1$H} NMR (149 MHz, THF-d$_8$, 298 K): δ 143.6 ($^1J_{Si, Sn}$ = 1208 Hz).

Absorption spectrum (THF): λmax (ε) = 342 (24201) and 263 (72193) nm.

Anal. Calcd for C$_{84}$H$_{118}$Br$_2$N$_{10}$Si$_2$Sn$_2$: C, 58.61; H, 6.91; N, 8.14. Found: C, 58.45; H, 6.83; N, 8.34.

**Preparation of Compound 4:** A toluene solution (10 mL) of MeOTf (83.5 mg, 0.5 mmol) was added to the solution of **2** (780.8 mg, 0.5 mmol) in fluorobenzene (10 mL) dropwise at 0 °C over a period of 10 min. The mixture was allowed to warm to room temperature. Then a fluorobenzene solution (10 mL) of Na[B(Ar$^F$)$_4$](Ar$^F$ = C$_6$H$_3$-3,5-(CF$_3$)$_2$) (443 mg, 0.5 mmol) was added dropwise over a period of 2 min. The solution was stirred at room temperature for 10 min and the solvent was removed under reduced pressure. Then the resulting solids were extracted with 30 mL toluene. After filtration and removal of the

solvent, the resulted purple solid was washed with 30 mL hexane to yield purple powder of **4** in 90 % yield. Single crystals suitable for X-ray diffraction studies were obtained by layering *n*-pentane on a fluorobenzene solution of **4** at room temperature.

$^1$H NMR (400 MHz, C$_6$D$_6$, 298 K): δ 8.44 (s, 8H, B(Ar$^F$)$_4$-Ar*H*), δ 7.71-7.69 (m, 6H, Ar*H*), 7.14-7.06 (m, 14H, Ar*H*), 7.03-6.96 (m, 4H, Ar*H*), 6.00 (s, 4H, NC*H*), 3.27-3.02 (m, 8H, C*H*(CH$_3$)$_2$), 1.52 (s, 3H, SnC*H*$_3$),1.40 (d, *J* = 6.4 Hz, 24H, CH(C*H*$_3$)$_2$), 1.13 (d, *J* = 6.8 Hz, 24H, CH(C*H*$_3$)$_2$), 0.74 (s, 36H, C(C*H*$_3$)$_3$). one peak for Ar*H* is overlapped with B(Ar$^F$)$_4$-Ar*H*. Several peaks for Ar*H* are overlapped with the solvent residual signal of C$_6$D$_6$.

$^{13}$C{$^1$H} NMR (101 MHz, C$_6$D$_6$, 298 K): δ 174.0 (s, N*C*N), 162.9 (q, $J_{C-B}$ = 50 Hz, B(Ar$^F$)$_4$-Ar-*C*), 147.5 (s, Ar*C*), 144.8 (s, Ar*C*), 135.5 (s, B(Ar$^F$)$_4$-Ar-*C*), 134.1 (s, Ar*C*), 131.0 (s, Ar*C*), 130.6 (s, Ar*C*), 130.4 (s, Ar*C*), 129.9 (m, B(Ar$^F$)$_4$-Ar-*C*), 129.3 (s, Ar*C*), 129.2 (s, Ar*C*), 125.7 (s, Ar*C*), 125.3 (q, $J_{C-F}$ = 274 Hz, B(Ar$^F$)$_4$-*C*F$_3$),124.8 (s, Ar*C*),118.1 (m, B(Ar$^F$)$_4$-Ar-*C*), 116.4 (s, N*C*H), 54.75 (*C*(CH$_3$)$_3$), 31.21 (*C*H(CH$_3$)$_2$), 28.93 (*C*H$_3$), 25.14(*C*H$_3$), 23.85 (*C*H$_3$). The resonances for Sn*C*H$_3$ can not be found because of topomerization. $^{19}$F NMR (377 MHz, C$_6$D$_6$, 298 K): δ −62.06 (s). $^{11}$B NMR (128 MHz, C$_6$D$_6$, 298 K):δ −11.09 (s). $^{29}$Si{$^1$H} NMR (79 MHz, C$_6$D$_6$, 298 K): no signal was observed due to the topomerization. $^{119}$Sn{$^1$H} NMR (149 MHz, C$_6$D$_6$, 298 K): no signal was observed due to the topomerization. Absorption spectrum (THF): λmax (ε) = 531 (8295) nm and 263 (90223) nm. Anal. Calcd for C$_{117}$H$_{133}$BF$_{24}$N$_{10}$Si$_2$Sn$_2$: C, 57.60; H, 5.49; N, 5.74. Found: C, 57.45; H, 5.61; N, 5.57.

**Preparation of Compound 5:**

**Method A:** A THF solution (1 mL) of KHB$^s$Bu$_3$ (0.5 mmol) was added to the solution of **4** (1088.2 mg, 0.5 mmol) in fluorobenzene (10 mL) dropwise. The solution was stirred at room temperature for 2 h and then the solvent was removed under reduced pressure. Then the resulting solids were extracted with a mixture of Et$_2$O (10 mL) and *n*-hexane (20 mL). After filtration and removal of the solvent, the resulted solid was dissolved in a mixture of Et$_2$O (10 mL) and *n*-hexane (5 mL). The suspension was filtered and the filtrate was stored at -30 °C for 2 days to yield the purple powder of **5** in 30% yield.

**Method B:** A toluene solution (10 mL) of MeOTf (83.5 mg, 0.5 mmol) was added to the solution of **2** (780.8 mg, 0.5 mmol) in fluorobenzene (10 mL) dropwise at 0 °C over a period of 10 min. The mixture was allowed to warm to room temperature and a THF solution (1 mL) of KHB$^s$Bu$_3$ (0.5 mmol) was added. The solution was stirred at room temperature for 2 h and then the solvent was removed under reduced pressure. Then the resulting solids were extracted with 30 mL Et$_2$O. After filtration and removal of the solvent, the resulted solid was dissolved in a mixture of Et$_2$O (10 mL) and n-hexane (5 mL). The suspension was filtered and the filtrate was stored at -30 °C for 2 days to yield the purple powder of **5** in 70% yield. Single crystals suitable for X-ray diffraction studies were obtained by evaporation of a Et$_2$O solution at room temperature.

$^1$H NMR (400 MHz, C$_6$D$_6$, 298 K): δ 8.12 (d, *J* = 7.6 Hz, 1H, Ar*H*), δ 7.95 (d, *J* = 6.8 Hz, 1H, Ar*H*), 7.42 (t, *J* = 6.8 Hz, 1H, Ar*H*), 7.31 (d, *J* = 6.8 Hz, 1H, Ar*H*), 7.21-7.19 (m, 2H, Ar*H*), 7.15-7.08 (m, 10H, Ar*H*), 6.96-7.92 (m, 2H, Ar*H*), 6.04 (s, 2H, NC*H*), 6.02 (s, 2H, NC*H*), 5.51 (s, 1H, NC*H*N), 3.57-3.48 (m, 4H, C*H*(CH$_3$)$_2$), 3.45-3.36 (m, 4H, C*H*(CH$_3$)$_2$), 1.65 (d, *J* = 6.8 Hz, 12H, CH(C*H*$_3$)$_2$), 1.25 (d, *J* = 6.8 Hz, 12H, CH(C*H*$_3$)$_2$), 1.16 (d, *J* = 6.8 Hz, 12H, CH(C*H*$_3$)$_2$), 0.98 (s, 18H, *t*Bu), 0.84 (s, 18H, C(C*H*$_3$)$_3$). Several peaks for Ar*H* are overlapped with the solvent residual signal of C$_6$D$_6$.one peak for Sn*C*H$_3$ is overlapped with the CH(C*H*$_3$)$_2$.

$^{13}$C{$^1$H} NMR (101 MHz, C$_6$D$_6$, 298 K): δ 171.4 (s, N*C*N), 151.8 (s, Ar*C*), 148.4 (s, Ar*C*),148.3 (s, Ar*C*), 145.6 (s, Ar*C*), 135.6 (s, Ar*C*), 135.3 (s, Ar*C*), 132.7 (s, Ar*C*), 131.4 (s, Ar*C*), 130.0 (s, Ar*C*), 129.8 (s, Ar*C*), 129.5 (s, Ar*C*), 127.5 (s, Ar*C*), 127.1 (s, Ar*C*), 127.0 (s, Ar*C*), 126.8 (s, Ar*C*), 124.6 (s, Ar*C*), 124.3 (s, Ar*C*), 116.0 (s, N*C*H), 115.2 (s, N*C*H), 72.33 (N*C*HN), 54.35 (*C*(CH$_3$)$_3$), 51.05 (*C*(CH$_3$)$_3$), 31.57 (*C*H(CH$_3$)$_2$), 31.44 (*C*H(CH$_3$)$_2$), 28.97 (*C*H$_3$), 28.84 (*C*H$_3$), 25.71 (*C*H$_3$), 25.67 (*C*H$_3$), 24.57 (*C*H$_3$), 24.29 (*C*H$_3$),

13.23 (Sn*C*H$_3$). one peak for Ar*C* is overlapped with the solvent residual signal of C$_6$D$_6$.

$^{29}$Si{$^1$H} NMR (79 MHz, C$_6$D$_6$, 298 K): δ −28.83 (s, *Si*MeSnSnSi) and −7.40 (s, SiMeSnSn*Si*).

$^{119}$Sn{$^1$H} NMR (149 MHz, C$_6$D$_6$ + THF, 298 K): δ 843.2 (s, Me*Sn*Sn) and 300.3 (s, MeSn*Sn*).

Absorption spectrum (THF): λmax (ε) = 396 (12115) nm, 533 (18078) nm, 264 (103043) nm and 224 (130650) nm.

Anal. Calcd for C$_{85}$H$_{122}$N$_{10}$Si$_2$Sn$_2$: C, 64.72; H, 7.80; N, 8.88. Found: C, 64.64; H, 7.66; N, 8.65.

**Reaction of 2 with DippN$_3$:** A Et$_2$O suspension (10 mL) of **2** (312 mg, 0.2 mmol) was added to the DippN$_3$ (163 mg, 0.8 mmol) in Et$_2$O (10 mL) over 2 min at room temperature leading to immediate gas evolution and a color change to yellow. The solvent was removed in vacuum and subsequent NMR analysis revealed clean formation of (SnNDipp)$_4$ and silaimine compound (from the reaction of LSi(NHI) (L= PhC(NtBu)$_2$) with the DippN$_3$: See SI) with the ratio 1:4. Characteristic NMR data of (SnNDipp)$_4$: $^1$H, $^{13}$C, and $^{119}$Sn NMR spectrum of (SnNDipp)$_4$ are in agreement with reported literature (see Supplementary Information).

$^1$H NMR (400 MHz, C$_6$D$_6$, 298 K): 1.29 (d, *J* = 6.4 Hz, 48H, CH(C*H*$_3$)$_2$). Several peaks for C*H*(CH$_3$)$_2$ and Ar*H* are partially overlapped with the peaks for the silaimine compound.

$^{13}$C{$^1$H} NMR (101 MHz, C$_6$D$_6$, 298 K): δ 148.1 (s, Ar*C*), 142.0 (s, Ar*C*), 125.4 (s, Ar*C*), 121.3 (s, Ar*C*), 30.59 (s, CH(*C*H$_3$)$_2$), 27.97 (s, *C*H(CH$_3$)$_2$).

$^{119}$Sn{$^1$H} NMR (149 MHz, C$_6$D$_6$, 298 K): δ 316.9.

$^{29}$Si{$^1$H} NMR (79 MHz, C$_6$D$_6$, 298 K): δ −105.57.

**Reaction of 2 with 4,6-di-*tert*-butyl-N-(2,6-diisopropylphenyl)-*o*-iminobenzoquinone (imQ):** A THF solution (5 mL) of **2** (312 mg, 0.2 mmol) was added to the **imQ** (455.5 mg, 1.2 mmol) in THF (10 mL) over 10 mins at room temperature and the color was changed to the light brown. The solvent was removed in vacuum and subsequent NMR analysis revealed the clean formation of the five-coordinate bis(amidophenolato)tin(IV) complex along with a by-product (from the reaction of LSi(NHI) (L = PhC(N$^t$Bu)$_2$) with the **imQ**: See Supplementary Information) and a little unknown compound with the ratio 1: 0.8: 0.2. Characteristic NMR data of the five-coordinate bis(amidophenolato)tin(IV) complex: $^1$H NMR spectrum of the compound are in agreement with reported literature (See Supplementary Information).

$^1$H NMR (400 MHz, toluene-d$_8$, 298 K): δ 6.85 (d, 2H, *J* = 2.3 Hz, Ar*H*), 6.29 (d, 2H, *J* = 2.1 Hz, Ar*H*, *J* (H-$^{117,119}$Sn) = 11.0 Hz), 4.05 (m, 2H$^α$, CH$_2$ group of THF), 3.76 (m, 2H$^α$, CH$_2$ group of THF), 3.40 (septet, 2H, 6.6 Hz, C*H*(CH$_3$)$_2$), 3.27 (septet, 2H, 6.4 Hz, C*H*(CH$_3$)$_2$), 1.34 (s, 18H, C(C*H*$_3$)$_3$), 1.19 (s, 18H, C(C*H*$_3$)$_3$), 1.03 (d, 6H, 6.9 Hz, CH(C*H*$_3$)$_2$). Several peaks for Ar*H*, three sets of peaks for CH(C*H*$_3$)$_2$ and one set of peaks for H$^β$ of THF are overlapped with the peaks from the reaction of LSi(NHI) (L= PhC(N$^t$Bu)$_2$) with the **imQ**.

$^{119}$Sn{$^1$H} NMR (149 MHz, toluene-d$_8$, 298 K): δ −294.8.

$^{29}$Si{$^1$H} NMR (79 MHz, toluene-d$_8$, 298 K): δ −115.81.

**Preparation of IPr → SnBr$_2$:** A toluene suspension (10 mL) of SnBr$_2$ (418 mg, 1.5 mmol) was added to the IPr (661 mg, 1.7 mmol) in toluene (10 mL) over 15 min at room temperature. After stirring for 48 h at this temperature, a white slurry was obtained. Then the solution was concentrated to 8 mL and the white precipitate was isolated by filtration to yield the **IPr → SnBr$_2$** in 85% yield. Single crystals suitable for X-ray diffraction studies were obtained by vapor diffusion of *n*-pentane into the THF solution at room temperature.

$^1$H NMR (400 MHz, C$_6$D$_6$, 298 K): δ 7.23 (t, *J* = 7.6 Hz, 2H, Ar*H*), 7.08 (d, *J* = 8.0 Hz, 4H, Ar*H*), 6.47 (s, 2H, NC*H*), 2.85-2.75 (m, 4H, C*H*(CH$_3$)$_2$), 1.42 (d, *J* = 6.8 Hz, 12H, CH(C*H*$_3$)$_2$), 0.98 (d, *J* = 6.8 Hz, 12H, CH(C*H*$_3$)$_2$).

$^{13}$C{$^1$H} NMR (101 MHz, C$_6$D$_6$, 298 K): δ 145.9 (s, Ar*C*), 133.7 (s, Ar*C*), 131.4 (s, N-*C*H-), 124.8 (s, Ar*C*), 124.6 (s, Ar*C*), 29.21 (s, *C*H(CH$_3$)$_2$), 25.84 (s, *C*H$_3$), 23.38(s, *C*H$_3$).

$^{119}$Sn{$^1$H} NMR (149 MHz, C$_6$D$_6$, 298 K): δ −21.3 (s).

Anal. Calcd for C$_{27}$H$_{36}$Br$_2$N$_2$Sn: C, 48.61; H, 5.44; N, 4.20. Found: C, 48.43; H, 5.36; N, 4.14.

## Data availability

All data generated or analyzed during this study are included in this manuscript (and its Supplementary Information). Details about materials and methods, experimental procedures, characterization data, and NMR spectra are available in the Supplementary Information. The structures of **1-5** in the solid state were determined by single-crystal X-ray diffraction studies and the crystallographic data have been deposited with the Cambridge Crystallographic Data Centre under nos. CCDC 2254817 (**1**), 2254818 (**2**), 2254819 (**3**), 2254820 (**4**), and 2254821 (**5**). Copies of the data can be obtained free of charge on application to CCDC. These data can be obtained free of charge from The Cambridge Crystallographic Data Centre via www.ccdc.cam.ac.uk/datarequest/cif. All data are also available from corresponding authors upon request. Coordinates of the optimized structures are present as source data. Source data are provided with this paper.

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

## Acknowledgements

We thank the financial support from the National Natural Science Foundation of China (Nos. 22071124, Z. M.; 22371130, Z. M.; 22188101, Z.M.; 22221002 Z.M.), Frontiers Science Center for New Organic Matter at Nankai University (C029215001), the Fundamental Research Funds for the Central Universities (Nos. 63206007, Z. M.), Nankai University, and Young Elite Scientists Sponsorship Program by Tianjin.

## Author contributions

Z.M. designed the project. S.D. and X.C. carried out the experiments. Z.M. and F.C. did the theoretical calculations. Z.M. analyzed the data and wrote the manuscript. S.D., H.R. and H.S. performed the single-crystal X-ray diffraction studies.

## Competing interests

The authors declare no competing interests.
