## [Peer Review File · Nature Communications]

A Silylene-Stabilized Ditin(0) Complex and Its Conversion to Methyliditin Cation and DistannavinylideneReviewers' Comments:

Reviewer #1:

Remarks to the Author:

The manuscript by Mo and coworker describe the synthesis of a ditin compound substituted with two N-heterocyclic silylenes (cpd 2). This compound is isoelectronic to a diphosphene with P=P double bond. Consequently, the bonding analysis of compound 2 reveals a Sn=Sn double bond with a lone pair at each Sn atom. The methylated species 4 has no resemblance to an alkynyl cation, the methylation occurs at the lone pair of the tin atom which is orthogonal to the n-system. The reaction of a hydride source with 4 leaves the multiple bond unaffected, very unusual for an distannynyl cation. The hydride adds to the NHSi indicating that the polarization of the NHSi-Sn(1)-bond in compound 4 (and 2) is (+)NHSi-Sn(-) and no arrow is needed. In summary, the synthesized compounds are interesting and well characterized. The computations are state of the art; however, I do not agree with the interpretation of the data. For the interpretation of the data, I recommend to compare the synthesized compounds with the isolobal phosphorous species and not with group 14 compounds. In summary, the data provided is fine, the interpretation is misleading and in the present version I do not recommend publication of this manuscript.

There are other mistakes that should be erased during revision of the manuscript.

Line 10. The structure of 2 was established by single crystal X-ray diffraction analysis (not by spectroscopic analysis).

Line 16. Are these compounds indeed "highly reactive". The reagents used are MeOTf and KHBBu₃ which are highly reactive reagents.

Line 26. The NHC stabilized diatomic main group compounds are not allotropes of these main group elements

Figure 1b. Please specify R₃Si.

Line 36. These compounds are not allotropes of the elements.

Line 44. Please use plural (...silylenesanalogues...)

Line 47. What is meant with a thermodynamic target? Please rephrase.

Line 53. Please define NHSi as LSiNHI

Lines 64-67. The involvement of NHSi coordinated Sn₂H₂ in the formation of 2 is a speculation, please omit.

NMR characterization of compound 2. Tin NMR data is missing, the SI informs that this is due to the high anisotropy of the chemical shift tensor. This might be true, however, give the spectral range which was investigated, either in the main text or in the SI material.

Line 85. ..."(2.7225(5) Å),6 but" please check or erase the "6"

Lines 85/86 actually the reported SnSn distance is longer than the quoted example.

Line 86: The compound should have a triple bond between the tin atoms

Line 87. 40 is the wrong reference.

Lines 92-93. The statement " The optimized structure of 2 is in excellent agreement " is completely unsupported. What is excellent (1% or 5% deviation)? Please provide a comparison of important computed and experimental structural data (SI material).

Lines 118-119: I do not agree that complex 4 with the composition [(NHSi)(Me)SnSn(NHSi)]⁺ (one Sn tricoordinated, the other dicoordinated) is an analogue of R-CC⁺ (one carbon dicoordinated the second monocoordinated). Please rephrase.

Line 129. ...methonium....

Lines 143- 152: Please give the WBI of the SnSn bond in compound 4 for comparison.

Lines 185 – 187. Sn₁ has a trigonal planer coordination environment and the coordination geometry of Sn₂ is V-shaped. Please rephrase.

Lines 196 The HOMO of 5 (not of 2)

References 38 and 40 are identical.

Reviewer #2:

Remarks to the Author:

Mo and co-workers present a manuscript about silylene stabilized ditin compounds. The presented chemistry is certainly of interest to the community of main group element chemists. This referee doubts that this ditin chemistry is of high interest to the broad readership of Nature communications and therefore suggests publication in a more suitable journal.

The missing ^{119}Sn NMR signal of 2 was only commented in the experimental section. This comment is missing in the manuscript.

The missing ^{119}Sn NMR signal of 4 was also only commented in the experimental section. Low temperature ^{119}Sn NMR measurements are needed. Because of the importance of the tin atoms in this chemistry more spectroscopic tin data are necessary.

To this referee it is very uncommon not to comment on the missing ^{119}Sn NMR data in a manuscript on tin chemistry.

The name distannynyl cation of 4 was derived from the term carbyne cation. To this referee the name distannynyl cation (4) suggests a triple bond could be discussed in the compound so designated. However, the cation 4 consist of a $\text{Sn}=\text{Sn}$ double bond with one tin atom alkylated and coordinated by a silylene, therefore the name distannynyl cation should not be used.

The hydride addition to compound 4 is interesting and the position of electrophilicity in cation 4 belongs to the NCN-ligand connected at the silicon. The authors did not comment on this reaction. A discussion should be added. What about the LUMO of cation 4?

Citations:

line 33, citation 33 is in wrong position.

line 40, citation 6 is wrong at this position, it should be 11, the Sn^2 -Jones paper

typos:

line 85, ... 2.7225(5) Å),6 but...

line 86, ..shorter than... should be longer than.

^{119}Sn NMR data should not be listed with two decimals.

Reviewer: 1

General Comments:

The manuscript by Mo and coworker describe the synthesis of a ditin compound substituted with two N-heterocyclic silylenes (cpd 2). This compound is isoelectronic to a diphosphene with P=P double bond. Consequently, the bonding analysis of compound 2 reveals a Sn=Sn double bond with a lone pair at each Sn atom. The methylated species 4 has no resemblance to an alkynyl cation, the methylation occurs at the lone pair of the tin atom which is orthogonal to the π -system. The reaction of a hydride source with 4 leaves the multiple bond unaffected, very unusual for a distannynyl cation. The hydride adds to the NHSi indicating that the polarization of the NHSi-Sn(1)-bond in compound 4 (and 2) is (+)NHSi-Sn(-) and no arrow is needed. In summary, the synthesized compounds are interesting and well characterized. The computations are state of the art; however, I do not agree with the interpretation of the data. For the interpretation of the data, I recommend to compare the synthesized compounds with the isolobal phosphorous species and not with group 14 compounds. In summary, the data provided is fine, the interpretation is misleading and in the present version I do not recommend publication of this manuscript. There are other mistakes that should be erased during revision of the manuscript.

Response:

We thank the reviewer for the high evaluation of the quality of this work and for the comments given below. In the light of the reviewers' comments, we have thoroughly revised the manuscript to improve this manuscript. According to the reviewer's suggestions, we have reformulated the tin compounds and modified the interpretation of the data. Compounds **2** and **4** are described as a ditin compound [$\{L(NHI)Si\}Sn=Sn\{Si(NHI)L\}$] and a NHSi-stabilized mixed-valent methylditin cation [$\{L(NHI)Si\}Sn(Me)=Sn\{Si(NHI)L\}[B(Ar^F)_4]$], respectively. The structural comparisons between compounds **4** and **5** with the corresponding isolobal multiple-bonded phosphorous compounds are included in the main text. The main comparisons are summarized as following:

“The methylditin cation **4** and distannavinylidene **5** are isolobal to multiple-bonded phosphorous compounds such as diphosphanyl cations (Fig. 1, **A**) and stannaphosphenes (Fig. 1, **B**). These structures display a trigonal planar geometry at the three-coordinated Sn and P atom. Recent quantum chemical calculations of $[Me_2P=PMe]^+$ and $[(ArDipp)\{CH_2P(CH_3)_2\}Sn=P(ArDipp)]$ ($Ar^{Dipp} = 2,6-(2,6-iPr-C_6H_3)-C_6H_3$) by the Filippou group (*J. Am. Chem. Soc.* **138**, 4589-4600 (2016)) and the Aldridge group (*J. Am. Chem. Soc.* **144**, 8908-8913 (2022)), respectively, reveal that they have similar frontier orbitals as those of **4** and **5**. The HOMO can be identified as the $\pi(E-E)$ orbital ($E = Sn$ or P) in all cases and the LUMO represents the $\pi^*(E-E)$ orbitals.”

Fig. 1. Isolobal phosphorous derivatives of compounds **4** and **5**, R = anionic ligands, Silyl = [Ph(H)C(NtBu)₂]Si.

In addition, the ability of the ditin(0) complex to transfer the Sn(0) to other substrates was investigated. The reactions of complex **2** with DippN₃ and iminobenzoquinone afforded the tetrameric tin imido complex [(SnNDipp)₄] and bis(amidophenolato)tin complex, highlighting its synthetic utility.

Comment 1:

Line 10. The structure of **2** was established by single crystal X-ray diffraction analysis (not by spectroscopic analysis).

Response:

We thank the reviewer for the comment. The description of spectroscopic analysis has been changed to “single crystal X-ray diffraction analysis” according to the reviewer’s suggestions.

Comment 2:

Line 16. Are these compounds indeed “highly reactive”. The reagents used are MeOTf and KHBBu₃ which are highly reactive reagents.

Response:

We thank the reviewer for the comment. We have modified the description.

Comment 3:

Line 26. The NHC stabilized diatomic main group compounds are not allotropes of these main group elements.

Response:

We thank the reviewer for the comment. The description of “serve as soluble allotropes” has been changed to “serve as precursors”.

Comment 4:

Figure 1b. Please specify R₃Si.

Response:

We thank the reviewer for the comment. We have redrawn the structures in the Fig. 1b

Comment 5:

Line 36. These compounds are not allotropes of the elements.

Response:

We thank the reviewer for the comment. The description of “diatomic allotropes” has been changed to “zero-valent diatomic main-group compounds”.

Comment 6:

Line 44. Please use plural (···silylenes ···analogues···)

Response:

We thank the reviewer for the careful examination. The “silylene” and “analogue” have been changed to “silylenes” and “analogues” according to the reviewer’s suggestions.

Comment 7:

Line 47. What is meant with a thermodynamic target? Please rephrase.

Response:

We thank the reviewer for the comment. We apologize for the inaccurate expression. The “thermodynamic and synthetic targets” has been changed to “synthetically achievable” according to the reviewer’s suggestions.

Comment 8:

Line 53. Please define NHSi as LSiNHI

Response:

We thank the reviewer for the comment. The “NHSi” has been changed to “LSiNHI” according to the reviewer’s suggestions.

Comment 9:

Lines 64-67. The involvement of NHSi coordinated Sn₂H₂ in the formation of 2 is a speculation, please omit.

Response:

We thank the reviewer for the comment. We have omitted the description about NHSi-coordinated Sn₂H₂ intermediate according to the reviewer’s suggestions.

Comment 10:

NMR characterization of compound 2. Tin NMR data is missing, the SI informs that this is due to the high anisotropy of the chemical shift tensor. This might be true, however, give the spectral range which was investigated, either in the main text or in the SI material.

Response:

We thank the reviewer for the comment. We apologize for missing the description of NMR data of compound 2 in the main text. The corresponding details has been added in the main text and SI material.

Comment 11:

Line 85. “...” (2.7225(5) Å),⁶ but “ please check or erase the “6 “

Response:

We thank the reviewer for the comment. “6” is the reference number. We apologize for citing the wrong literature. The “6” has been changed to the “11”.

Comment 12:

Lines 85/86 actually the reported SnSn distance is longer than the quoted example.

Response:

We thank the reviewer for the comment. We apologize for the wrong description. We have changed “shorter” to “longer” according to your suggestion.

Comment 13:

Line 86: The compound should have a triple bond between the tin atoms

Response:

We thank the reviewer for the comment. The double bond ([Ar'Sn=SnAr']) in the manuscript have been changed to triple bond ([Ar'Sn≡SnAr']) according to your suggestion.

Comment 14:

Line 87. 40 is the wrong reference.

Response:

We thank the reviewer for the comment. We apologize for quoting the wrong literature. The valid literature has been cited in the reference according to your suggestion.

Comment 15:

Lines 92-93. The statement “The optimized structure of 2 is in excellent agreement ...” is completely unsupported. What is excellent (1% or 5% deviation)? Please provide a comparison of important computed and experimental structural data (SI material).

Response:

We thank the reviewer for the comment. The description has been changed to “the optimized

structure of **2** agrees well with the X-ray derived structure”. We have added a comparison of the computed and experimental structural data in the Supplementary Information (See Table S3).

Comment 16:

Lines 118-119: I do not agree that complex **4** with the composition [(NHSi)(Me)SnSn(NHSi)]⁺ (one Sn tricoordinated, the other dicoordinated) is an analogue of R-CC⁺ (one carbon dicoordinated the second monocoordinated). Please rephrase.

Response:

We thank the reviewer for the comment. We have reformulated the tin compounds and modified the interpretation of the data. Compound **4** is described as a NHSi-stabilized mixed-valent methyliditin cation $[\{L(NHI)Si\}Sn(Me)=Sn\{Si(NHI)L\}][B(Ar^F)_4]$.

Comment 17:

Line 129. ...methonium...

Response:

We thank the reviewer for the comment. The “distannamethonium ion” has been changed to “distannamethonium ion” according to your suggestion.

Comment 18:

Lines 143- 152: Please give the WBI of the SnSn bond in compound **4** for comparison.

Response:

We thank the reviewer for the comment. We have described the WBI of the Sn=Sn bond of **4** in the revised text. “The WBI of the Sn–Sn bond in **4** (1.84) is larger than that of **2** (1.77) and is in line with its enhanced double bond character.”

Comment 19:

Lines 185-187. Sn1 has a trigonal planer coordination environment and the coordination geometry of Sn2 is V-shaped. Please rephrase.

Response:

We thank the reviewer for the comment. We have corrected the description according to your suggestion. “The Sn2 atom features a V-shaped geometry with a Sn1–Sn2–Si2 angle of 97.79(9)°.”

Comment 20:

Lines 196. The HOMO of **5** (not of **2**).

Response:

We thank the reviewer for the comment. “The HOMO of 2” has been changed to “The HOMO of 5” according to your suggestion.

Comment 21:

References 38 and 40 are identical.

Response:

We thank the reviewer for the comment. We apologize for quoting the wrong literature (40). The valid literature has been cited in the reference according to your suggestion.

Reviewer: 2

General Comments:

Mo and co-workers present a manuscript about silylene stabilized ditin compounds. The presented chemistry is certainly of interest to the community of main group element chemists. This referee doubts that this ditin chemistry is of high interest to the broad readership of Nature communications and therefore suggests publication in a more suitable journal.

Response:

We thank the reviewer for the high evaluation of the quality of this work and for the comments given below. In the light of the reviewers’ comments, we have thoroughly revised the manuscript to improve this manuscript. We have further studied the ability of the ditin(0) complex to transfer the Sn(0) to other substrates. The reactions of complex **2** with DippN₃ and iminobenzoquinone afforded the tetrameric tin imido complex [(SnNDipp)₄] and bis(amidophenolato)tin complex, highlighting its synthetic utility. These new results have been included in the revised manuscript.

The synthesis of molecules that featuring main-group elements in unusual oxidation states is a primary pursuit of main-group chemistry, which expands our understanding of bonding and electronic structure and serves as the foundation for the construction of long sought after molecules that would otherwise remain inaccessible. In this contribution we demonstrate **the synthesis, structural characterization and reactivity of a silylene-stabilized ditin(0) complex**. The key features of this work are briefly summarized below:

1) Synthesis and characterization of a silylene-stabilized ditin(0) complex. The chemistry of zero-valent main-group element complexes emerged recently and has attracted increasing interest due to their unusual electronic structures and their synthetic potential. Due to their low electronegativity and the less effective pπ–pπ orbital overlap of tin atom, isolation of tin(0) complexes remains a great challenge. *We demonstrates the utility of a strongly electron-donating N-heterocyclic imino substituted silylene in stabilization of highly reactive ditin(0) complexes, providing a new strategy to probe the exciting chemistry of highly reactive tin(0) complexes. In terms of utility, the ditin(0) complex is capable of transferring the Sn(0) fragment to other*

substrates.

2) Developing a strategy to access unprecedented ditin cation via alkylation of the ditin(0) complex. Ditin cation are intriguing targets as they possess two highly reactive functional groups including a main-group element cation and a multiple bond. However, the synthesis of such species is extremely challenging as the lack of a general method. *Addition of a small electrophile (Me^+) to the silylene-stabilized ditin(0) afforded a methyl-ditin cation featuring mixed-valent Sn(II) and Sn(0) centers, which undergoes topomerization in solution through a reversible 1,2-Me migration along a Sn=Sn bond.*

3) Synthesis and characterization of the first distannavinylidene. According to the computational findings, the positive charge in ditin cation is mainly at the three-coordinate Sn atom. Thus, the addition of nucleophiles to it might give the tin analogues of vinylidene, which remains exclusive in spite of considerable efforts have been made to synthesize heavier analogues of vinylidene compounds over the past decade. *The ditin cation reacts with $KBH(sec-Bu)_3$, where hydride attacks the amidinate ring to form the an NHSi-stabilized distannavinylidene, providing another example for the exceptional ability of NHSis to stabilize low-valent tin centers in unusual bonding environments.*

As such, we strongly believe this paper will be of broad general interest, not only to those working in the highly topical field of main group chemistry, but also to colleagues working in the broader fields of synthetic chemistry.

Comment 1:

The missing ^{119}Sn NMR signal of 2 was only commented in the experimental section. This comment is missing in the manuscript.

Response:

We thank the reviewer for the comment. We apologize for not commenting the missing ^{119}Sn NMR signal of 2 in the manuscript and the corresponding description has been added.

Comment 2:

The missing ^{119}Sn NMR signal of 4 was also only commented in the experimental section. Low temperature ^{119}Sn NMR measurements are needed. Because of the importance of the tin atoms in this chemistry more spectroscopic tin data are necessary.

Response:

We thank the reviewer for the comment. The low temperature ^{119}Sn NMR measurements of 4 has been performed according to your suggestion. The related data has been summarized in the Fig. 5 of SI Material and discussed in the main text.

“The ^{119}Sn NMR spectrum of 4 recorded at 213 K shows two signals at δ 633.4 and 297.9 ppm,

which correspond to the tri-coordinate Sn(II) atom and the terminal two-coordinated Sn(0) atom, respectively. The resonance for the Sn(0) atom is shifted downfield relative to that of the bis(NHSi)-stabilized zero-valent tin complex ($\delta = -1147.2$ ppm)".

Comment 3:

The name distannynyl cation of **4** was derived from the term carbyne cation. To this referee the name distannynyl cation (**4**) suggests a triple bond could be discussed in the compound so designated. However, the cation **4** consist of a Sn=Sn double bond with one tin atom alkylated and coordinated by a silylene, therefore the name distannynyl cation should not be used.

Response:

We thank the reviewer for the comment. We have reformulated the tin compounds and modified the interpretation of the data. Compound **4** is described as a NHSi-stabilized mixed-valent methylditin cation $[\{L(NH)Si\}Sn(Me)=Sn\{Si(NH)L\}][B(Ar^F)_4]$.

Comment 4:

The hydride addition to compound **4** is interesting and the position of electrophilicity in cation **4** belongs to the NCSi-ligand connected at the silicon. The authors did not comment on this reaction. A discussion should be added. What about the LUMO of cation **4**?

Response:

We thank the reviewer for the comment. According to your suggestion, we have added the comments on the formation of **5**.

"Distannavinylidene **5** might be formed via two different routes. One direct pathway could proceed by the addition of the hydride to the PhC site of amidinate as the LUMO+1 of **4** is comprised of the π^* orbitals localized on the amidinate. An alternative pathway involves the addition of the hydride to the Sn1 atom followed by the cleavage of the Sn–H bond by NHSi in a cooperative fashion, as has been recently described in the cooperative B–H bond activation by the amidinate-stabilized silylene (*Chem. Commun.* **55**, 3536-3539 (2019))."

DFT calculations indicate that the LUMO of **4** represents the $\pi^*(Sn-Sn)$ orbitals and the positive charge in **4** is mainly at the Sn1 atom. We deduced that the second pathway might be more reasonable.

Comment 5:

line 33, citation 33 is in wrong position.

Response:

We thank the reviewer for the comment. We apologize for citing the wrong literature. The citation 33 (references) has been modified to the correct literature according to your suggestion.

Comment 6:

line 40, citation 6 is wrong at this position, it should be 11, the Sn2-Jones paper

Response:

We thank the reviewer for the comment. We apologize for citing the wrong literature. The “6” has been changed to the “11” according to your suggestion.

Comment 7:

line 85, ... 2.7225(5) Å,6 but...

line 86, ..shorter than... should be longer than.

Response:

We thank the reviewer for the comment. The “shorter than” has been changed to “longer than” according to your suggestion.

Comment 8:

¹¹⁹Sn NMR data should not be listed with two decimals.

Response:

We thank the reviewer for the comment. The ¹¹⁹Sn NMR data in the manuscript and SI material has been revised and listed with one decimal.

We would like to take this chance to thank you and the reviewers very much for editing and reviewing this manuscript. Your valuable comments have made a much better presentation of this paper. I trust these revisions fairly address the points raised by the reviewers and sincerely hope that this revised manuscript is now acceptable for publication. Thank you!

Yours sincerely,

Zhenbo Mo

Reviewers' Comments:

Reviewer #2:

Remarks to the Author:

The author has improved the manuscript and my criticisms and concerns have been answered. I still think that the chemistry presented is of high level but very specific for a Journal like Nat Comm. Nevertheless, I do not disagree with publication in Nat. Comm

Reviewer #3:

Remarks to the Author:

[Note from the Editor: Reviewer #3 was invited to assess the response given to reviewer #1 who was not able to look over the revision again.]

The paper by Du and co-workers introduces some Sn=Sn compounds with Sn(0) that have been characterised with multiple techniques and computational methods. Although the paper has been peer-reviewed before, and I think that the authors have addressed most of the issues raised by reviewer 1, I do believe that some corrections need to be addressed before the paper can be accepted for publication in nature communications.

1. I wouldn't say the field is in its infancy. I would suggest something more rigorous, such as early stages.

2. The authors do some TD-DFT calculations, which are discussed in the SI. The results reported in there are very confusing. First, I do not know exactly how this calculations are done, because there are the labels TD-HF and TD-DFT. There is no such thing as TD-HF. Also, the transitions are reported with number, but I'm assuming that this calculations are done in a restricted framework, thus making the energy of the alpha and beta orbitals equal. If that is the case, there should be at least two equal contributions in theTDDFT orbital section (one for the alpha and one for the beta). Also, the contributions do not seem very large (0.68), and it jumps from transitions 1-3 to 22. I'm assuming this are the transitions with oscillator strengths larger than 0.0, but it's not clear at all. would suggest the authors to review and clarify this section.

3. Indeed, the computed and experimental structure do agree. I would add the overall error in the main manuscript.

4. It seems to me that there is a mix of basis sets in the methodology. Sometimes the method is PBE0-D3(BJ)/Def2-TZVP and some others PBE0-D3(BJ)/SVP. Why there is such distinction? When is one or the other used?. For the type of calculations done in this job, I would use at least the Def2-TZVP basis set, but perhaps the authors have some comparison between both methods? Can they provide with a clear comment on why and how is each method used and for what purposes?

5. The fact that all angles add to 359.95 doesn't mean it's a triangle (T-shape angles also add to 360).

6. The last section is called "discussion" but it should be "conclusions", I guess?

7. By looking at the structures, I was wondering if any agnostic interaction between the lone pairs and the H atoms is observed.

8. In figure 9, as well as in the SI, the NBO notation (LP, BD, ..) is used but not defined. Please clarify this in the text.

After addressing all these points, the paper can be reevaluated and accepted for publication

Reviewer: 2

General Comments:

The author has improved the manuscript and my criticisms and concerns have been answered. I still think that the chemistry presented is of high level but very specific for a Journal like Nat Comm. Nevertheless, I do not disagree with publication in Nat. Comm.

Response:

We would like to thank the referee again for taking the time to review our manuscript and for your appreciation on our work. The raised comments are all of great importance to our article and have contributed a lot to improve the quality of our article.

Reviewer: 3

General Comments:

The paper by Du and co-workers introduces some Sn=Sn compounds with Sn(0) that have been characterized with multiple techniques and computational methods. Although the paper has been peer-reviewed before, and I think that the authors have addressed most of the issues raised by reviewer 1, I do believe that some corrections need to be addressed before the paper can be accepted for publication in nature communications.

Response:

We thank the reviewer for the high evaluation of the quality of this work and for the comments given below. In the light of the reviewers' comments, we have thoroughly revised the manuscript to improve this manuscript.

Comment 1:

I wouldn't say the field is in its infancy. I would suggest something more rigorous, such as early stages.

Response:

We thank the reviewer for the comment. The description of "in its infancy" has been changed to "in the early stages" in manuscript according to the reviewer's suggestions.

Comment 2:

The authors do some TD-DFT calculations, which are discussed in the SI. The results reported in there are very confusing. First, I do not know exactly how this calculations are done, because there are the labels TD-HF and TD-DFT. There is no such thing as TD-HF. Also, the transitions are reported with number, but I'm assuming that this calculations are done in a restricted framework, thus making the energy of the alpha and beta orbitals equal. If that is the case, there should be at least two equal contributions in the TDDFT orbital section (one for the alpha and one for the beta).

Also, the contributions do not seem very large (0.68), and it jumps from transitions 1-3 to 22. I'm assuming this are the transitions with oscillator strengths larger than 0.0, but it's not clear at all. would suggest the authors to review and clarify this section.

Response:

We thank the reviewer for the comment. We apologize for the inaccurate expression about the TD-DFT calculations in the supplementary information.

The method used for TD-DFT calculations is "pbe1pbe def2svp TD(nstates=30) scrf(SMD,solvent=thf)". The output files from TD-DFT calculations report the excitation energies and oscillator strength for each excited state. The descriptions in SI are the output information which shows the excitation energies and oscillator strengths for the selected excited states including the first excited state and the ones with large oscillator strengths. To avoid the ambiguity, the first excited state and the excited state with the largest oscillator strength are retained in the revised SI.

As described by the reviewer, TD-DFT calculations are done in a restricted framework, the excited states for the alpha and beta orbitals are completely consistent, and only the information for the alpha part is given in the output file. For example, here is the output section for the second excited state of ditin(0) **2** from the TD-DFT calculation:

```
Excited State 2: Singlet-A 2.0471 eV 605.66 nm f=0.1236 <S**2>=0.000
      382 -> 383          0.69178
```

The transition constant for 382 → 383 is 0.69178. Therefore, the contribution of 382 → 383 to this excitation is $(0.69178^2) \times 100 = 95.7\%$.

TD-DFT calculation also outputs the total Energy for the first excited state. Take ditin(0) **2** for example, $E(\text{TD-HF}/\text{TD-DFT}) = -4819.66387962$. The total energy is the sum of the energy of the ground state of the molecule $E(\text{TD-HF})$ plus the excitation energy $E(\text{TD-DFT})$. The value labeled TD-HF is actually the result of DFT, not the result of Hartree-Fock.

Comment 3:

Indeed, the computed and experimental structure do agree. I would add the overall error in the main manuscript.

Response:

We thank the reviewer for the comment. The overall error has been added in the main text according to the reviewer's suggestions.

Comment 4:

It seems to me that there is a mix of basis sets in the methodology. Sometimes the method is PBE0-D3(BJ)/Def2-TZVP and some others PBE0-D3(BJ)/SVP. Why there is such distinction? When is one or the other used? For the type of calculations done in this job, I would use at least the Def2-TZVP basis set, but perhaps the authors have some comparison between both methods? Can they provide with a clear comment on why and how is each method used and for what purposes?

Response:

We thank the reviewer for the comment. The molecules in this paper consists of more than 200 atoms, which makes the calculation quite time-consuming. Therefore, we use a mix of basis sets for structure optimization and single point energy calculation. Geometry optimizations were performed in the gas phase with the PBE0/Def2SVP level of theory and the Vibrational frequency calculations were carried out at the same level of theory as the geometry optimizations to provide the thermal corrections for Gibbs free energy determinations. The single-point calculations of the optimized geometries were performed with PBE0 functional with Grimme's D3BJ dispersion correction and triple-zeta quality Def2TZVP basis set.

According to the reviewer's suggestion, geometry optimizations were recalculated with Def2TZVP basis set. The data have been updated in the revised manuscript and SI. Compared with the previous method, the structures optimized with PBE0-D3(BJ)/Def2TZVP showed little difference in bond lengths.

Comment 5:

The fact that all angles add to 359.95 doesn't mean it's a triangle (T-shape angles also add to 360).

Response:

We thank the reviewer for the comment. We apologize for the inaccurate expression and the original description has been corrected according to the reviewer's suggestions.

Comment 6:

The last section is called "discussion" but it should be "conclusions", I guess?

Response:

We thank the reviewer for the comment. The description of "discussion" has been changed to "conclusions" in manuscript according to the reviewer's suggestions.

Comment 7:

By looking at the structures, I was wondering if any agnostic interaction between the lone pairs and the H atoms is observed.

Response:

We thank the reviewer for the comment. As for compound **2**, **4** and **5**, The shortest distances between the Sn atom with lone pairs and the H atoms are 3.333 Å, 3.226 Å and 3.201 Å respectively. These distances indicate that there is no obvious interaction between the lone pairs of Sn and the H atoms.

Comment 8:

In figure 9, as well as in the SI, the NBO notation (LP, BD, ..) is used but not defined. Please clarify

this in the text.

Response:

We thank the reviewer for the comment. We have clarified all the NBO notation in the main text and SI.

We would like to take this chance to thank you and the reviewers very much for editing and reviewing this manuscript. Your valuable comments have made a much better presentation of this paper. I trust these revisions fairly address the points raised by the reviewers and sincerely hope that this revised manuscript is now acceptable for publication. Thank you!

Yours sincerely,

Zhenbo Mo

Reviewers' Comments:

Reviewer #3:

Remarks to the Author:

In this revision, the authors have addressed all my concerns, and upgraded significantly the computational part, which is my field of expertise. There are of course potential source of discussion, but as for now, the calculations reported in this work are sound.

As far as I'm concerned, the paper can be published in it's current form.

Congratulations!

Reviewers: 3

General Comments:

In this revision, the authors have addressed all my concerns, and upgraded significantly the computational part, which is my field of expertise. There are of course potential source of discussion, but as for now, the calculations reported in this work are sound. As far as I'm concerned, the paper can be published in it's current form. Congratulations !

Response:

We thank the reviewer for the high evaluation of the quality of this work and for the comments.

We would like to take this chance to thank you and the reviewers very much for editing and reviewing this manuscript. Your valuable comments have made a much better presentation of this paper. I trust these revisions fairly address the points raised by the reviewers and sincerely hope that this revised manuscript is now acceptable for publication. Thank you!

Yours sincerely,

Zhenbo Mo